# The Need to Bridge the Gap between Research on Children's Rights and Parenting Styles: Authoritative/Democratic Style as an Acultural Model for the Child's Well-Being

**Galym Zhussipbek** [1,*] **and Zhanar Nagayeva** [2]

[1] Department of Social Sciences, Suleyman Demirel Atindagi Universitet, Kaskelen, Almaty 040900, Kazakhstan
[2] Independent Researcher, Almaty 050000, Kazakhstan
*   Correspondence: galym.zhussipbek@sdu.edu.kz

**Abstract:** The United Nations Convention on the Rights of the Child contains specific provisions on parent–child relations and parenting, but these provisions can be described as elusive. Furthermore, the Convention does not explicitly specify a children's rights-friendly parenting style. On the other hand, there is a disconnect between research on children's rights and parenting styles. Based on the insights of the meta-theoretical critical realist approach, this paper argues that universal human flourishing is inconceivable without the development of a children's rights-friendly parenting style. It is argued that the Convention's provisions on parent–child relations can be adapted to the perceptions of average parents, especially living in paternalistic societies, by adapting the conceptualizations of parenting styles developed by Baumrind and Lakoff. Overall, research on children's rights, supported by literature on children's rights-friendly parenting, can show that children's rights do not alienate parental rights and responsibilities. Instead, children's rights give appropriate direction to parental authority and responsibility to realize the child's well-being.

**Keywords:** children's rights; Convention on the Rights of the Child; parenting style; child's well-being; Baumrind; Lakoff; humanistic education; authoritative parenting; authoritarian parenting; paternalism; Eurasian country

## 1. Introduction

A recent report from the World Health Organization shows that in the past year, up to one billion children aged 2–17 years worldwide have experienced physical, sexual, or emotional violence or neglect, which can impact their lifelong health and well-being (Violence Against Children 2022).

Alderson (2017), a pioneer in the field of children's rights, points out that adults are much better able to defend their rights in public and in private than children. Moreover, criticism of children's rights has millions of hits on Google searches compared to criticism of women's rights or human rights in general. It can be argued that children's rights are among the most misunderstood categories of human rights (Alderson 2017). Therefore, it is troubling, but logical, that, at the individual country level, particularly in post-socialist Eurasian countries, conspiracy theories about children's rights disseminated as part of the information war have led to a strong backlash from parent groups against legislative reforms aimed at tightening domestic violence laws. In Kazakhstan, for example, this parental opposition has resulted in the parliament halting the drafting of a new law prohibiting all types of violence against children (first author's personal communication with the head of the anti-torture NGO, Tatyana Chernobyl in Almaty, 9 December 2022). Discourse analysis of the prejudices and misinformation about children's rights prevalent in Kazakhstan and other Eurasian countries shows that parents' rights and children's rights are intentionally and falsely presented in dichotomized relationships (Yessimkhanov 2020, 2021).

A child's well-being is the central concept of contemporary legal and welfare systems and is inextricably linked to the realization of children's rights (Dolan et al. 2020). Thus, a child's well-being and children's rights are linked to parental education. Parenting is about parental authority and parent–child relations. Therefore, the UN Convention on the Rights of the Child (UNCRC) contains specific provisions that regulate parental authority and parent–child relations. Articles 3 (1), 5, 18, and 19 refer directly to parent–child relations. The main concepts used to determine parenting in the UNCRC are the observance of the best interests of the child, consideration of the evolving capacities of the child, parental guidance, and care. However, the UNCRC does not elaborate on what type of parenting is appropriate to a child's evolving capacities and consistent with the best interests of the child, which are defined from the perspective of the child, not the adult. Moreover, as Freeman (2007a, p. 69) notes, the UNCRC does not provide clear instructions on how to raise children.

The United Nations Committee on the Rights of the Child (the Committee) has published the comments and decisions on the UNCRC provisions regulating parenting to address the ambiguity of concepts. However, the Committee's decisions are not binding, so these commentaries may not have a serious impact given the persistence and reproduction of paternalistic patterns and norms, especially in authoritarian countries and paternalistic societies.

Recent research on children's rights (see, e.g., Tobin and Varadan 2019) shows that authoritarian models of parenting only harm the evolving capacities of a child. However, the link to children's rights principles is generally poorly conceptualized in research on parenting styles. Similarly, children's rights research does not openly discuss the factors that are essential to a child's happiness, health, and maturity that can be found in studies of parenting styles and in humanistic education, such as Montessori education philosophy. Therefore, we argue that there is a disconnect, first, between research on children's rights and parenting styles and second, between research on children's rights and humanistic education (which is beyond the scope of this article).

We premise this article on the ideas of meta-theoretical critical realism, which aims to avoid the extremes of positivistic and postmodernist approaches (Bhaskar 2016). Critical realism is based on reflexivity; it aims to understand social phenomena, not just describe them (Bhaskar 2016). Critical realism defends interdisciplinarity; moreover, it proposes a post-disciplinary approach in social sciences (Sayer 2011). We accept that ontological realism, a main assumption of critical realism (Bhaskar 2016), is a necessary conceptual tool to defend the universality of human rights and, in particular, the universality of children's rights. Moreover, critical realism seeks to ask the so-called 'big questions' of human life, such as "What is justice? What is morality? How to achieve universal human flourishing?" (Sayer 2011). By engaging in the discussion about children's rights-friendly parenting, this paper aims to conceptualize the grounds of child flourishing.

Consistent with the main assumptions of critical realism, this research is conceptual–normative. We base our theoretical assumptions on the practical experience of the first author, who worked with students for ten semesters (from spring 2016 to spring 2022) as part of the human rights course at one of the universities in Almaty, Kazakhstan, and on the interactions of the second author (a certified Montessori early childhood educator) with parents of young children in Almaty. We use the evidence from our practical experience as anecdotal only, as we were unable to obtain written consent from the students and parents to use their data in our research

The main argument of this article is that, in realizing the child's well-being (or child flourishing), it is necessary to present children's-rights-friendly parenting styles more explicitly in research on children's rights and parenting. The point is to link the UNCRC (normative children's rights principles) to the practical realm, which is about the realization of the child's well-being. Moreover, this explicit connection will make the provisions of the UNCRC that regulate the relationship between parents and children understandable and actionable for average parents. In particular, parents and educators in authoritarian

countries and paternalistic societies, whose parenting styles, thought patterns, and practices are influenced and shaped by their social environment, need more actionable children's-rights-oriented information.

One of the classifications of parenting styles widely used in the democratic educational context was conceptualized by Baumrind and further developed by Maccoby and Martin. The American political philosopher and cognitive linguist Lakoff has also developed a valuable conceptualization of parenting styles that are consistent with major contemporary political ideologies. We cannot make these conceptualizations of parenting styles absolute. However, based on our life and professional experience in an authoritarian country and paternalistic society, we argue that these conceptualizations of parenting style can provide good guidance for understanding the UNCRC provisions on parenting and the parent–child relationship. Overall, we argue that the concrete parenting styles can be seen as helpful conceptual tools in clarifying the meaning of the UNCRC provisions, particularly in identifying the styles, models, and practices that are consistent with the principles of children's rights, and understood through the prism of children's developmental needs but not through the prism of adults.

The remainder of the article is organized as follows. First, it addresses the need to accept the universality of the child's innate nature and child's well-being. Then, the concepts of "social determination" of parenting and family support, the UNCRC's provisions on the parent–child relations, their elusiveness and the paternalistic challenge to children's rights are discussed. We then address the need for comparable parenting styles and provide a brief description of parenting styles according to Baumrind and Lakoff's conceptualizations and discuss them through the prism of children's rights. Then we provide some insights from our work experience in a paternalistic societal context, showing how explicit parenting styles are necessary to develop children's rights. Finally, we seek to substantiate our views on the universality of the authoritative/democratic parenting style, which can be described as children's rights-friendly and necessary to achieve the child's well-being.

This paper is intended to contribute to the process of the social construction of children's rights. It aims to help counter relativist challenges to the universality of human rights by underpinning the universality of children's rights. It also aims to contribute to research that examines the contexts in which children's rights and UNCRC are applied.

## 2. The Universality of the Child's Innate Nature, the Child's Well-Being, and Critical Realism

Children's rights are human rights. Therefore, the characteristics of children's rights are to be universal, inherent, indivisible, and inalienable. These characteristics require the acceptance of the universality of the child's innate nature and basic needs, which underlie the humanistic philosophy of education and at the same time are the result of it (Montessori 2014; Korczak 2016). Critical realism accepts ontological objectivity or ontological realism, which is by no means synonymous with uniformity and the denial of the extraordinary diversity of human life (Sayer 2011). Acceptance of ontological realism is necessary to conceptualize the axiological principle of the contemporary concept of human rights—the respect for the equal ontological or moral dignity of human beings regardless of gender, origin, age, abilities, beliefs, and lifestyles.

We argue that the acceptance of ontological objectivity or ontological realism is consistent with the idea of the universality of the child's innate nature and basic needs. The assumption that childhood is different and socially constructed because children experience different childhoods (adults create these different childhoods) (Peleg 2012, p. 223) does not negate the universality of the child's innate nature. Therefore, it is not correct to argue that children in different cultural contexts supposedly have "different" psychologies and basic needs, whether in Western, Asian, or post-socialist Eurasian contexts.

The universality of the innate nature of the child suggests that children's needs and the laws of child development are functional and "acultural", beyond cultural contexts. Therefore, we argue that there are universal principles for understanding the universal basic

needs of a child that are necessary for the child's well-being (Zhussipbek and Nagayeva, Forthcoming).

Following the "big question" raised by critical realism, "How can universal human flourishing be achieved?" we argue that universal human flourishing is inconceivable without realizing the well-being of the child. Child well-being is a multifaceted and complex phenomenon. As Dolan et al. (2020) note, it is used as an overarching concept that refers to the right to grow, learn, achieve, feel safe and secure, to have a voice, to participate and influence, and to have positive personal and social relationships. We conceptualize children's well-being as they grow up happy, mature, and healthy, which can be achieved through the development of children's rights-friendly parenting. In summary, child's well-being and children's rights are inextricably linked (Dolan et al. 2020). Child-rights-friendly parenting needs to be given a higher profile in the research and literature on children's rights and parenting.

## 3. The UNCRC and Parenting Styles

### 3.1. Social Determination of Parenting and Family Support as a Right of the Child

Critical realist scholars emphasize that humans are by nature not only capable and active beings but also needy, dependent, and vulnerable (Sayer 2011). This human condition is evident in the study of childhood, which cannot be abstracted from parenting styles, the nature of parental authority, and parent–child relations. Parenting, which creates a kind of world around the child, has a decisive influence on the child's life.

In researching parenting, it is important to examine not only specific practices but also broader patterns or the "parenting milieu", represented as parenting styles. Parenting style can be understood as a normal variation in parents' attempts to control and socialize their children (Baumrind 1991). Parenting, of course, includes nonverbal children. However, as Cole-Albäck (2020) points out, research on children's rights has focused primarily on verbal children, even though many works on early childhood show that early childhood experiences are critical to adult well-being. For example, adverse experiences in early childhood can lead to long-term detrimental effects across the lifespan (see, e.g., Tarsha and Narvaez 2022). Moreover, early brain development is the most important predictor of health and well-being in a person's lifetime (Fitzgerald 2016; O'Connor et al. 2016); even the child's academic trajectory begins in early childhood, not in school (Williams et al. 2016)

Montessori (1995) emphasized that the child has an absorbent mind, which unconsciously and with unlimited speed soaks up and absorbs all impressions from the environment. The impressions become a part of the child's psyche, shape the child's personality, and remain with them throughout life. From birth, children observe and imitate the movements of others around them. As "strangers in a strange land", children absorb an enormous amount of information long before birth; therefore, an orderly and healthy environment is critical for children. Parenting style has such an impact on a person's life that, in cases of traumatic parental behavior, so-called "aftercare" may be needed by victimized children when they are no longer "chronological children" (Freeman 2007a, p. 68).

Following the logic of Yamin (2008), who emphasizes the "social determination" of health, we also use the concept of the "social determination" of parenting instead of "social determinants" of parenting. Similarly to health, the "social determination of parenting" means that parenting is produced in the social, political, historical, and economic contexts in which humans live. In other words, the social and cultural environment created by formal and informal norms and beliefs in parenting (which can either catalyze or seriously harm the child's development and well-being) can be tentatively depicted as the "social determination of parenting". Overall, there is a direct nexus between social and cultural norms and parenting styles, which inevitably influences the implementation of children's rights.

On the other hand, the "social determination" of parenting" is about class inequalities, employment, education, and housing patterns, which, for example, have overarching importance to health (Yamin 2008). Critical scholars believe that psycho-social factors

directly influence early child development by shaping the environment in which children live (Marmot 2017). Moreover, children's health and developmental outcomes are related to the social and economic conditions of their families; the higher up the socioeconomic spectrum, the better the outcomes (Moore et al. 2015). Similarly, persistent poverty has far-reaching and long-lasting negative effects on young children's brain development and health (Moore et al. 2015).

Leyendecker et al. (2006) point out that children from disadvantaged groups not only perform worse in school but are also more likely to experience harsh, inconsistent parenting and be exposed to chronic stressors. Therefore, Leyendecker et al. (2006) argue that socio-economic status matters within the context of parenting and child development. There is a direct nexus between persistent or temporary poverty and the detrimental effects associated thereby (Leyendecker et al. 2006), including the development of negative parenting.

While we cannot deny that economic and social factors, structurally embedded in the political, socio-economic, cultural, and educational systems, play a role in shaping parenting, we also cannot absolutize this connection. Montessori, for example, originally developed her humanistic educational philosophy for the children of the most disadvantaged and marginalized social groups. She emphasized that the most valuable thing parents and generally adults can give children is their warmth, love, and attention, which do not depend on their social status and educational level (Ferreira 2014).

As Dolan et al. (2020) point out, the family is the most important institution in society to protect, nurture, and ensure the general well-being of the child; therefore, family support should be considered a fundamental right of the child. Building family capacity is an integral part of efforts to realize children's rights and the child's well-being of children (Dolan et al. 2020). However, "the century-old idea that the best way to help parents is to disseminate child development information should include a major focus on the social contexts of parenting" (Leyendecker et al. 2006).

### 3.2. Parenting in the UNCRC: The Normative Foundations of Children's Rights-Friendly Parenting

The UNCRC contains provisions on the parent–child relations, the limits of parental authority, and parental responsibilities. These provisions can be categorized as follows: the right to parental guidance and direction (Article 5); the role of parents and other legal guardians, who are primarily responsible for the child (Article 18(1)); the right of the child to an adequate standard of living (Article 27); the right of the child to be free from violence (Article 19); the obligation of states to assist parents in fulfilling their child-rearing responsibilities (Article 18(2)) and to take measures to assist parents and others to provide an adequate standard of living for the child (Article 27(3)); the responsibility of states to raise awareness about children's rights and to disseminate the principles of the UNCRC (Article 42).

The most important article is Article 5, the thrust of which is about positioning parents and caregivers as participants in the child's life but not as determiners (Kamchedzera 2012, p. 14), as can be understood in the paternalistic context. This article prescribes the right of children to appropriate guidance and direction from parents so that they can exercise their rights, rather than the right of parents to have their rights respected by the state in relation to their upbringing (Tobin and Varadan 2019, p. 161). For Kamchedzera (2012, para. 22), innovation in this article stems from three foundations: the need to take into account the evolving capacities of the child in providing appropriate guidance and direction, the centrality of children's rights rather than a "welfarist" approach, and the need to delineate the scope of parental authority and discretion.

Overall, the importance of the UNCRC stems from its revolutionary attempt to firmly establish the child as a rights-bearer in international law and to draw a clear distinction between parental authority over the child and the child's right to enjoy their own rights (Kamchedzera 2012, p. 13). Tobin and Varadan (2019, p. 181) make it even clearer by pointing out that the UNCRC shows that the child is neither a possession of the parents nor

of the state nor simply an object of care. Articles 5, 18, and 12 together emphasize the need for parenting styles, care, and instruction that respect the child as an individual person and as a rights-bearer.

The right of the child to grow up in a humane, non-violent environment is part of Article 5 (parental guidance), Article 7 (parental care), Article 3 (the best interests), Article 19 (the child's right to be free from violence), and an adequate standard of living for the child (Article 27). In addition, the UNCRC provides a normative framework that psychologists can use as a guide when advocating for children's needs (Melton 2005).

One of the concepts used in the UNCRC to define parent–child relations is the evolving capacities of the child. Recognition of the child's evolving capacities implies protection of the child from arbitrary and unrestricted control by the family (Sutherland 2020, p. 451). Specifically, Article 5 requires parents to recognize the evolving capacities of the child (Tobin and Varadan 2019, p. 161). It is unique in international law in that it links the child's evolving capacities with appropriate guidance and direction (Kamchedzera 2012, p. 6). The concept of evolving capacities underscores the child's right to parental guidance and direction. Specific provisions of the convention, such as family reunification, also imply that parents have primary responsibility for the child's development in accordance with the child's evolving capacities (Nowak 2005, p. 7).

Tobin and Varadan (2019, p. 181) point out that the principle of evolving capacities underlies the positive concept of discipline in the context of protection. In a non-authoritarian parenting context, the concept of positive discipline is fundamental. The Committee uses the principle of evolving capacities as a conceptual tool to address the issues concerning children's rights, such as violence against children and corporal punishment (Tobin and Varadan 2019, p. 181). Recognizing the child's emerging agency is critical to determining the limits of parental authority, which is not properly understood in an authoritarian, paternalistic context. Overall, authors who analyze parenting according to the provisions of the UNCRC emphasize the importance of non-authoritarian parenting, which is premised either on the idea of evolving capacities (Tobin and Varadan 2019; Sutherland 2020; Varadan 2019) or of dignified life (Kamchedzera 2012), the inherent dignity of human being to which every human being from birth is entitled (Alderson 2015, 2019; Freeman 2007b).

*3.3. Elusiveness of the UNCRC Provisions on Parenting*

Despite the normative underpinnings and specific literature on the children's rights parenting noted above, we argue that a more thorough overview of the UNCRC and disengagement from the unproblematized approach to it can show that the Convention's provisions do not contain overt definitions and references to parenting styles that are consistent with children's rights principles and values. Instead, we can argue about the elusiveness of the UNCRC's provisions on parenting. Furthermore, the UNCRC contains contradictions (Quennerstedt et al. 2018), which is understandable given that comprehensive international agreements such as the UNCRC were the result of intense negotiation and mutual compromise by sides that could have opposing views and sought to minimize disagreements that arose during the drafting process. Quennerstedt et al. (2018) remind us that, "Attention to conflicting aspects within the convention is limited, and instead, the assumption that the convention represents an international consensus on the meaning of children's human rights seems to be widespread in policy and academic work". Mower (1997, p. 4) also points out that some states have chosen to become party to the UNCRC in order to avoid the appearance that they do not care about children, even though they are aware of the many loopholes the document contains.

Logically, the UNCRC does not openly explain what is meant by the evolving capacities of the child. That the Committee rarely refers to the obligations contained in Article 5 could be explained by the elusiveness of the concept of the evolving capacities of the child (see, for example, Sutherland 2020). Parker (1994, p. 27) also argues that it is impossible to speak of only one standard of the best interests of the child; instead, the differences in the formulation of this principle may be significant. In short, in the relevant provisions of

the UNCRC, the meaning of the principle of the best interests of the child and the content of the child's evolving capacities remain ambiguous. Therefore, it can be argued that the scope and limits of parental authority are unclear and that the Convention does not openly define the model or style of parenting that can be described as children's-rights-friendly.

On the other hand, children's rights literature contends that, while the meaning of concepts (such as best interests, appropriate direction, and guidance) may remain elusive and contentious, this does not mean that their contours are without limits (Sutherland 2020). Nor is the claim that the provisions governing parent–child relationships are obviously contradictory plausible. Tobin and Varadan (2019, p. 184) indicate that the new model of parenting envisioned by the UNCRC shows "no tolerance for authoritarian models of parenting where there is a little or no respect for the evolving capacities of children, the human dignity of children".

Nevertheless, we argue that even the relative elusiveness of the concepts (used to define the parent–child relations and parenting) can set the stage for paternalistic interpretations that are inconsistent with the UNCRC's teleological, contextual, and goal-oriented understanding. In this context, Freeman (2007a, p. 50) suggests that the UNCRC does not define the principle of the best interests of the child, nor does it elaborate on it as much as it should. Furthermore, to Freeman (2007a, p. 50), the principle of the best interests of the child appears paternalistic because it is viewed from an adult perspective. This emphasis would have been weakened if Article 3(1) had referred to the views of the child. Based on this criticism, Freeman (2007a, p. 69) argues that the UNCRC does not provide clear guidance on how to raise children.

While the Convention emphasizes that the best interests of children are the primary concern of parents, the rights and duties of parents are not defined in Article 3(2) and thus nowhere in the Convention. For example, although parental responsibility is mentioned in Article 18, it is not overtly defined, although reference is made to "upbringing and development" (Freeman 2007a, p. 69).

The basic concepts of protection and care, which are crucial to defining parental responsibility to the child, also defy clear definition. While the UNCRC asserts that states parties undertake to provide such protection and care as is necessary for the child's well-being, it does not specify the perspective from which viewpoint "necessary" is considered (Freeman 2007a, p. 68). The Committee, scholars, and experts need to do more to "determine the measures required of states to effectively respect, protect, and fulfill the right of children to receive parental direction and guidance consistent with their evolving capacities" (Tobin and Varadan 2019, p. 84).

The Committee has made an important contribution to the interpretation of the provisions of the UNCRC. The Committee's role is crucial in explaining the meaning of the various provisions in light of the general purposes, principles, and values of the Convention. In the absence of a judicial individual right of appeal, the Committee's General Comments and Concluding Observations are the primary jurisprudential source on the UNCRC (Lundy 2007). Moreover, the UNCRC is socially constructed by the Committee. Therefore, in order to eliminate ambiguity in the terms used in the provisions regulating parenting and parent–child relations, the Committee has issued comments and decisions, including those contained in the evaluations and recommendations on states' periodic reports. For example, some insights into the understanding of the best interests of the child can be gained by examining the committee's reports (Freeman 2007a, p. 50).

The Committee condemns corporal punishment, whether in the home, at school, or in institutions, as not in the child's best interests and calls on states that have ratified the UNCRC to prohibit corporal punishment through legislative reforms and other necessary measures. (Freeman 2007a, p. 69). However, due to the non-binding nature of its decisions, and given the persistence, reproduction, and reinvention of paternalistic principles and values, particularly in authoritarian countries, these comments may not have serious effect.

Recent work on children's rights (see, e.g., Tobin and Varadan 2019; Varadan 2019) provides discussion of the committee's evolving capacities. However, the elusiveness of

concepts and principles related to parenting in the UNCRC poses a serious challenge, evidenced by a low level of understanding of how the UNCRC operates on parenting issues (e.g., the content of positive discipline, a foundational idea of non-authoritarian parenting, remains unclear).

We argue that the elusiveness of the concepts of the best interests, the evolving capacities of the child, parental guidance, and care paves the way for different varieties of overt or latent paternalistic interpretations. Thus, different "local" understandings of the best interests of the child can be observed worldwide, which are permeated by paternalistic positions.

## 4. Paternalistic Challenge and the Need for Concrete and Comparative Parenting Styles

Scholars in the field of children's rights (Cole-Albäck 2015; Reynaert et al. 2012) remind us that, even in democratic countries, professionals working in education rarely use the UNCRC as a frame of reference for their practice because they lack knowledge and understanding of how this legal document relates to educational practice. Moreover, in a democratic country, children may complain that they do not have a voice (see, e.g., Lundy 2007). Nonetheless, pro-children's rights values such as respect for human personality and personal boundaries and the rule of law are institutionalized and practiced by default by a relative majority in a democratic social context. Thus, we argue that children in authoritarian and paternalistic societies (where authoritarian and paternalistic formal and informal institutions directly influence and shape parenting styles and practices) need more protection from the abusive and inherently paternalistic norms and practices internalized by their parents and other adults. Moreover, we argue that the principles and values well-established in democratic societies regarding the treatment of children in the family environment and in education need special clarification in a non-democratic/authoritarian and paternalistic context characterized by authoritarian formal and informal institutions.

We believe that, in the research on parenting styles, the connection to the principles of children's rights should be made more explicit, because in the academic discussion on parenting, the connection to the core values and principles of children's rights is missing, while the emphasis is on the success and happiness of a child. On the other hand, research on children's rights lacks a focus on a child's success, maturity, and happiness. The connection between the two: respecting children's rights in parenting and raising a happy, successful, and healthy child is especially important for parents living in authoritarian countries with a relatively low record of democratic values.

We agree with Tobin and Varadan (2019, p. 161) that the values underlying Article 5 cannot be considered new or particularly problematic in modern democratic societies. Nevertheless, these values may be seen as "foreign" and harmful in societies that may have inherited strongly authoritarian child-rearing practices or that are experiencing the rise of conservative paternalistic norms. For example, explaining the limits of parental authority and a broad interpretation of Articles 5 and 18 are challenging. In other words, although a thorough reading of the UNCRC in light of the Committee's and experts' in the field interpretations implies that a child has the right to grow up in a humane environment, many parents unfamiliar with this type of literature may have difficulty grasping children's rights-friendly parenting without tangible, concrete models.

We should take into account the fact that authoritarian parenting (low in responsiveness to a child's needs) arises in the social, political, historical, and economic contexts in which people live, which can be called the "social determination of authoritarian parenting" (Zhussipbek and Nagayeva, Forthcoming). Therefore, for parents and educators in authoritarian countries and paternalistic societies, due to the strong influence of paternalistic norms, the language of children's rights is the only way to enhance and expand the position of children in society. However, this language should be understandable; it should be adapted to the perception of an average parent.

Parents socialized under the influence of authoritarian, paternalistic norms tend to accept that paternalistic attitudes toward the child (which they describe as a centuries-old legacy of folk wisdom or an indispensable part of their cultural heritage) "serve" the child's best interests and evolving capacities of the child. The decisions of international bodies such as the Committee and the provisions of UNCRC may have little effect on them unless they are provided with concrete examples from education and psychology of the long-term negative effects of authoritarian (and long-term positive effects of children's-rights-friendly) parenting styles. Overall, explicitly defined parenting styles can be used to align the UNCRC with the perceptions of the average parent, including those who live in paternalistic societies or who are influenced by paternalistic norms.

## 5. Parenting Styles According to the Conceptualizations of Baumrind and Lakoff

Baumrind is one of the most influential scholars in the field of parenting, whose typological approach to parenting is recognized as foundational. Originally, Baumrind conceptualized three major parenting styles or models of parental control: authoritative (democratic), authoritarian, and permissive (Baumrind 1966). These parenting configurations were the prototypes that exemplified the characteristic features and explicit descriptions of parental behaviors. In general, Baumrind (1971) laid the foundation for the configurational approach to conceptualizing parenting styles, viewing them as configurations of different values, attitudes, and patterns of behavior. Maccoby and Martin (1983) later developed this typology and conceptualization further, emphasizing that a parenting style is based on two dimensions of parental behavior: parental demandingness and responsiveness.

While demandingness means to what extent parents want to control their children's behavior and demand their maturity, responsiveness means to what degree parents feel and show affinity to their children. More specifically, responsiveness (or parental warmth and supportiveness) refers to the extent to which parents foster individuality, self-regulation, and self-assertion by being attuned, supportive, and acquiescent to children's needs and requests. It includes warmth, autonomy support, and reasoned communication (Baumrind 2005, pp. 61–62; Baumrind 1991, p. 62). Darling and Steinberg (1993, p. 492) formulate demandingness as the "parent's willingness to act as a socializing agent" and responsiveness as "the parent's recognition of the child's individuality".

Based on the interaction of demandingness and responsiveness, Maccoby and Martin (1983) classified four parenting styles: authoritative (with high demandingness and high responsiveness), authoritarian (with high demandingness and low responsiveness); indulgent or permissive (with low demandingness and high responsiveness), and neglectful or uninvolved (with low demandingness and low responsiveness). This typology was later reformulated and expanded by Baumrind. Disengaged parenting, which denotes the parenting style low on both dimensions—demandingness and responsiveness (Baumrind 2005, p. 62), was added. The later categorizations of parenting styles became more sophisticated. The following typologies of the parents were added to the initial types: parents who are average on one or both dimensions, such as "directive" ("authoritarian–directive"—the parents who are imbalanced by being lowly responsive, highly intrusive, and highly demanding, "non-authoritarian–directive"—the parents who are moderately responsive, and moderately or low intrusive); "good enough" (parents are moderately demanding and moderately responsive) (Baumrind 2005, p. 62; Criss and Larzelere 2013, p. 1).

Parenting styles directly affect child's happiness, character development, health, maturity, and overall development. Parenting styles were formulated as empirical descriptions of how parents are labeled respectively mature, dysphoric, disaffiliated, or immature, and differed among themselves on responsiveness and demandingness variables. For example, the parents of mature children possessed the qualities of authoritative/democratic parents (who are highly responsive, low intrusive but also highly demanding) (Baumrind 2005, p. 62), whereas the parents of dysphoric or disaffiliated children had authoritarian tendencies. In addition, permissive parents tended to have immature children (Baumrind 1966).

According to decades of research, authoritative/democratic parenting style synthesizing nurturance and discipline is accepted as the most effective at fostering high achievement, emotional adjustment, self-reliance, self-discipline, and social confidence in children and adolescents. Adolescents whose parents use the authoritative/democratic parenting style perform better in school; they are more self-reliant and less likely to report feeling depressed and involved in the delinquent activity (Steinberg et al. 1995, p. 443). Hence, this style can be accepted as optimal for child development (Larzelere et al. 2013). However, Steinberg et al. (1995, p. 425) urge that, during middle and late adolescence, the influence of parents is likely to be moderated by forces outside the family, including the adolescent's peer group and the broader community in which the family lives.

American cognitive linguist and philosopher Lakoff (2009) conceptualized two major models of parenting, the Nurturant Parent (democratic) and the Strict Parent (conservative, paternalistic), to explain political and social processes. Lakoff's Nurturant Parent model is based on positive discipline; it arises from the child's developing sense of care and responsibility (Lakoff 2004, p. 4; Lakoff 2009, p. 81). This model is inherently empathetic and democratic. The basic idea is that "morality is caring". Nurturant parents demonstrate a high sense of responsibility while maintaining boundaries. In other words, nurturant parents are responsive to children's needs and opinions, and these parents set standards for children's behavior through their reasoning with them (Lakoff 2009, pp. 105, 177). This model is consistent with Baumrind's authoritative/democratic style.

The Strict (conservative) Parent model is based on negative discipline; it is inherently hierarchical, deficient in empathy, and undemocratic. The basic idea is that "morality is obedience to authority". Parents show little responsiveness while making high demands. The idea that "morality is strength" fits well with the primary idea that morality is obedience to authority. Authority requires strength to command obedience (Lakoff 2009, p. 105). This is consistent with Baumrind's authoritarian style.

## 6. Baumrind's and Lakoff's Parenting Styles through the Prism of Children's Rights

We believe that the authoritative/democratic parenting style (Baumrind) and the Nurturant Parent model (Lakoff) tentatively represent children's rights-friendly parenting. The dialog and high responsiveness of parents indicate that parents who use this style recognize children as rights holders and allow them to develop fully.

The high responsiveness of parents and their emotional and physiological attachment to the child is a prerequisite for taking into account the child's views (according to age) in all decisions that affect the child. This is an essential condition for safeguarding the child's best interests and ensuring the child's right to development. The invaluable advantage of authoritative/democratic education is also reflected in the preservation of the freedom of expression and the right to participation. Peleg (2012, p. 195) emphasizes that ensuring freedoms in the present will enable children to realize their potential and lead a life worth living in the future. For this reason, children's voices and opinions should not be silenced or dismissed but rather amplified.

Children's rights require that children be encouraged to participate in decisions that affect them. Because children are the actors and rights holders, they can participate, and their participation is a fundamental human right (Freeman 2007a). As Lundy (2007) points out, it is "difficult to imagine egregious breaches of children's rights in situations where they have been fully and effectively involved in determining the outcome of the decisions which affect them". Freedom of expression and the right to participate are also inseparable from the principle of respecting the evolving capacities of the child. Child-centered humanistic education assumes that children grow in independence when they are given freedom and clear boundaries (Ferreira 2014). In other words, if children are allowed freedom and independence within the wise limits set by nurturant parents, these children's abilities will develop in such a way that they become active, independent, and maximize their potential (Montessori 1995).

Understanding Article 5 as providing direction and guidance on a continuum (Sutherland 2020, p. 463) can confirm Baumrind's authoritative model. In our view, the scope of the authoritative/democratic parenting style and the Nurturant Parent model can also be extended to specific language communication with children from the earliest days. The enormous advantage of the authoritative and nurturant parenting style built on dialogue with children can also be sought in the phenomenon of language nurturing (Fitzgerald 2016; O'Connor et al. 2016). According to research, not only is parental warmth necessary to raise healthy and happy children, but parents should also talk to their children from their earliest days.

However, in the authoritarian parenting style, parental control tends to be excessive, even oppressive. Authoritarian parents tend to demand blind obedience. Communication with children is underdeveloped or nonexistent. Consequently, authoritarian parents are not responsive to their children's needs. Children rarely have a say in decisions that affect them. However, the evolving capacities of the child require the protection of the child from unrestricted adult control (Sutherland 2020). Intrusiveness and excessive control and restrictions imposed by authoritarian parents do not allow children to develop.

With authoritarian parenting, rules are strict and leave no room for interpretation; punishments are also applied quickly. In addition, authoritarian parents prefer strict, harsh discipline and use punishment frequently. They do not even shy away from punishing their children with corporal punishment. The authoritarian model is inherently paternalistic and violates the most important principles of children's rights. Moreover, authoritarian parenting, especially through its inherent tendency to physical and psychological violence, harms children and builds barriers between them and adults. Authoritarian parenting cannot be a basis for raising a happy and healthy child and protecting children's rights.

Harsh parenting methods used by authoritarian parents can lead to adverse childhood experiences (ACEs) that cause serious health problems and unhealthy habits (such as drug addiction, alcoholism, and others) in teens and later in adulthood (About the CDC-Kaiser ACE Study 2020; Harris 2014). In other words, authoritarian child-rearing methods, especially corporal punishments, constitute a traumatic experience. Moreover, the scholars of children's rights emphasize that denying the child's right to be free from corporal punishment means undermining the child's status as a human being and the integrity of children's human rights (Freeman 2007b, p. 7).

Other parenting styles, such as permissive and neglectful (uninvolved or disengaged), also contradict the principles of children's rights. Permissive parents fail to fulfill their parental duties and responsibilities, primarily by setting examples and establishing rules that impede the child's right to develop. Neglectful parents are indifferent to the needs of their children and do not participate in their lives. Neglected children may show deficits in many areas, including health and physical development, intellectual and cognitive growth, emotional and psychological well-being, and social and behavioral development (Fallon et al. 2014, p. 707). Neglectful parenting leads to problematic behaviors in adolescents, such as delinquency, depression, and discipline problems. Neglectful parenting also causes ACEs, as the parent's withdrawal from the child's life is very traumatic for the child. Given the broad definition of violence against children, efforts to end violence against children should not be limited to physical, psychological, and sexual violence, they should also include all forms of neglect (Doek 2019, p. 24).

## 7. Explicit Parenting Styles and Challenges of Paternalism: Insights from a Paternalistic Eurasian Society

Our work experience in a paternalistic societal context demonstrates the need for explicit parenting styles to develop children's rights. In the curriculum of the first author's human rights course, topics related to children's rights, including an analysis of the provisions of the UNCRC, child–parent relations, and adolescent rights, accounted for one-third of the semester's materials. His teaching experience has shown that Baumrind's and Lakoff's parenting styles can serve as good conceptual tools to explain the main prin-

ciples of children's rights, e.g., to demonstrate the limits of parental authority, the broad scope of children's rights, and the value of respecting children's rights to achieve the child's well-being. In addition, Lakoff's approach could help explain the role and influence of parenting on individual, social, and political life. Using the comparative and practical parenting styles in the study of children's rights could help students learn that authoritarian/paternalistic and neglectful parenting styles are deficient in parental responsiveness and traumatic to children's well-being and their relationships with parents or caregivers.

The ten semesters long interaction with the students shows that the students who performed well in writing and oral presentation and exhibited a commitment to civic activism came from families in which the parents used styles similar to authoritative/democratic parenting. The practical experience of the second author shows that the parents of children closer to the state of the so-called "normalized child" (a child who is helpful, shows concentration, self-discipline, love of work, sociability, and calmness in the Montessori kindergarten environment) (Ferreira 2014) are characterized by high responsiveness and warmth towards the child and prefer positive discipline to negative discipline. On the other hand, a more detailed analysis of the parenting style of parents whose children show different types of deviance in the preschool environment revealed that these parents either use harsh parenting methods or systematically neglect and pay little attention to the child.

Based on our experience in a society where conservative paternalistic norms or the patterns of authority and subordination ("culture of dominance") are common, we can claim that there is a widespread confusion between strictness and discipline when discipline is understood in a negative terms, namely, to be based on fear of punishment by "superiors". In other words, in a paternalistic environment, demandingness and strict parenting are readily confused. Parental strictness, and even harsh parenting methods, may be perceived as "understandable demandingness." However, "demandingness refers to the parent's willingness to act as a socializing agent, whereas responsiveness refers to the parent's recognition of the child's individuality" (Darling and Steinberg 1993, p. 492).

Our experience also shows that, in a paternalistic society, the principles of the best interests of the child and guidance according to evolving capacities of the child can be defined and understood solely from the adult perspective. Moreover, our interactions with students reveal that some inherently authoritarian and traumatic for the child informal parenting norms can be accepted as normal, even desirable. Most disturbingly, adults who have adopted paternalistic norms in child-rearing tend to accept that paternalistic attitudes toward the child (which may be considered a part of the "cultural" heritage) "serve" the child's best interests and evolving capacities of the child. Moreover, these types of paternalistic perceptions and traumatic norms can be popularized by educators, bloggers, and opinion leaders; some of them may even promote the false perceptions that are inherently harmful to a child's development.

A paternalistic society's stereotypes of child-friendly parenting can be neutralized by highlighting the differences between authoritative (or Nurturant) and authoritarian (or Strict) parenting styles and their impact on personal health, well-being, and social life. Overall, our experience can support claims about the universality of the authoritative/democratic parenting style, which can be described as children's rights-friendly and necessary to achieve the child's well-being, which we discuss below.

## 8. Authoritative–Democratic Style as an 'Acultural' Framework for the Child's Well-Being

Based on our long experience of living and working in educational institutions in a paternalistic Eurasian society, we can argue that positive behaviors such as good mental health, maturity, good self-esteem, social bonds, and high academic achievement are the result of the authoritative/democratic parenting style (Baumrind) or the Nurturant Parent model (Lakoff), which are a synthesis of caring and positive discipline. This fact is already well-established in the Western context. For example, Baumrind (1971) and Maccoby and Martin (1983) emphasized in their early work that the authoritative/democratic style in

democratic Western society is associated with the positive and mature behaviors mentioned above. Later research (e.g., Steinberg et al. 1995) also confirms that authoritative/democratic parenting styles are considered the most effective in developing positive and mature behaviors in children and adolescents.

We argue that a broader interpretation of the UNCRC, supported by the Committee's comments, demonstrates that child-centered authoritative/democratic parenting styles, characterized by a high degree of responsiveness to the child's needs, provide the foundation for universal principles of children's rights-friendly parenting that is open to local enrichments. We cannot deny that authoritarian, paternalistic parenting styles characterized by low responsiveness to the child's needs may have different meanings for different cultural groups (Ang and Goh 2006, pp. 5–6). Authoritarian parenting styles, however, which have typically been rooted in social and cultural norms, may pose a relativistic challenge to the universality of children's rights. Strictness toward children, negative discipline, humiliation, and dominance are harmful and dangerous to a child's health and well-being (e.g., Hogye et al. 2022), as well as to the well-being of society (see, e.g., Lakoff 2004, 2009; Montessori 1995, 2014). The reasons for the systemic violations and the consequently slow development of children's rights should be sought in the "social determination" of parenting with low responsiveness to the needs of the child, such as authoritarian and neglectful (see, for example, Zhussipbek and Nagayeva, Forthcoming).

A combination of nurturance and control is optimal in all cultural settings (Sorkhabi and Mandara 2013). In general, adolescents do better when their parents are authoritative/democratic, regardless of their racial, ethnic, or social background or the marital status of the parents (Steinberg 1990). This finding has been confirmed in samples from countries around the world that have extreme differences in their value systems, such as China, Pakistan, Hong Kong, Scotland, Australia, and Argentina (Baumrind 2013, p. 12). The first author's teaching experience shows that, after studying children's rights and parenting styles, most students admitted that they would prefer for their childhood the parenting styles that Baumrind describes as authoritative (democratic) and Lakoff as nurturing. Students are "yesterday's children", basically, it is the children's perceptions that matter (Sorkhabi and Mandara 2013).

In other words, the available research and our experience suggest that the styles of parenting—the authoritative style (Baumrind) or Nurturant model (Lakoff)—are applicable across cultures and can be effective among different cultural groups, such as Central Asian. The so-called "culture claims" should not be used to defend "cultural authenticity or uniqueness". In essence, not only are children's rights universal, but culture is also dynamic; it is not homogeneous but heterogeneous and hybrid. Acceptance of the universal values and principles of human rights is not cultural appropriation in the negative sense (Mende 2021).

Research on children's rights, supported by literature on parenting styles, is crucial to show parents that children's rights do not alienate parental rights and responsibilities (in a paternalistic context, children's rights and parental responsibilities are presented in dichotomous relationship), but that children's rights direct parental authority and responsibility in the right direction to realize the child's well-being or to raise a healthy, mature, and happy child (the goal that paternalistic parents also strive for but by holding on to the ideas and using the methods that contradict the values and principles of children's rights).

Ultimately, the authoritative parenting (Baumrind) or Nurturant Parent (Lakoff) model, which can be seen as an implementable children's rights-friendly parenting style, is not necessarily Western or European. They may also be applicable in a non-Western context, such as Eurasian society. The UNCRC elaborates on the state's responsibilities in securing support to parents to fulfill their parental responsibilities in the best interests of the child (Dolan et al. 2020, p. 9).

One of the best forms of parental support is to promote competent parenting, which should focus on children's rights-friendly parenting styles and methods, high in demandingness and responsiveness to the child's needs. The establishment and functioning of

the European Family Support Network (EurofamNet), a pan-European family support network, can be seen as a good example of the practical implementation of the provisions of the UNCRC. EurofamNet intends to inform family policy and practice with the ultimate goal of ensuring children's rights and families' well-being, including by promoting positive parenting built on children's rights. EurofamNet seeks to help develop and implement European approaches to family policy and family support that are more grounded in children's rights (Churchill et al. 2021).

Research on the development of democratic institutions has overlooked the role and importance of children's rights and parenting. However, the most effective antidote to political and social authoritarianism can only emerge in the field of parenting and education (Semenets 2020).

## 9. Conclusions

The provisions of the UNCRC, the most important document in the field of protection of children's rights, on parent–child relations and parenting can be described as elusive. Therefore, it is difficult to answer the questions "What does a children's rights-friendly parenting style look like?", "What are the characteristics of children's rights-friendly parenting?", and "What are the limits of parental authority?" without resorting to specific literature. The elusiveness of the key principles of children's rights, such as the best interests of the child, evolving capacities, protection, and care, can pave the way for paternalistic and authoritarian interpretations of child–parent relationships. The Committee's explanatory opinions on the UNCRC and its comments on states' periodic reports are crucial to understanding the Convention. However, an average parent may not be familiar with the Committee's decisions, especially if they live in authoritarian countries and paternalistic societies. Moreover, the official bodies of non-democratic countries may react to the Committee's criticisms with reluctance, to say the least, and repeatedly postpone the implementation of its decisions. This may be the case, for example, in some post-socialist Eurasian countries.

On the other hand, despite well-established research on parenting, there is still a deficiency of a clearly emphasized connection between the observance of children's rights in parenting and realization of the child's well-being. In addition, research on children's rights and parenting lacks the focus on early childhood development that humanistic education approaches and philosophies such as the Montessori philosophy can support.

The use of Baumrind's authoritative parenting style and Lakoff's Nurturant Parent model as conceptual tools in explaining how to realize the child's well-being by invoking children's rights can be seen as navigating an average parent's understanding of the provisions of the UNCRC. These parents may have little familiarity with the specific normative literature. Moreover, children's rights-friendly conceptualizations of parenting can help build more compelling arguments that authoritarian–paternalistic and neglectful parenting styles are incompatible with children's rights. Thus, the use of explicit models and parenting styles is necessary to explain the characteristics of children's rights-friendly parenting, especially among parents socialized under the influence of authoritarian, paternalistic formal and informal norms.

We assume that debates about the nature of the relationship between the notion of universality of human rights, including children's rights, and different cultural contexts will not be resolved at a certain level in the foreseeable future. Moreover, the goal of practical implementation of the universality of children's rights will provoke the assertion of various relativistic positions (Alston 1994, p. 16). Therefore, the development of children's rights should be supported by research on parenting, which is premised on child psychology and humanistic education that can be understood and adapted by the average parent.

The child's well-being and children's rights are inextricably linked (Dolan et al. 2020). Therefore, this paper argues that universal human flourishing is inconceivable without the realization of the child's well-being, which is conditioned by developing children's rights-friendly parenting. The authoritative/democratic parenting style can be seen as

an 'acultural', children's rights friendly parenting for raising happy, mature, and healthy children in all cultural contexts.

Since parents and families are the best child protectors if appropriately supported, the right of the child to family support becomes a prerequisite for the full realization of children's rights (Dolan et al. 2020, p. 9). However, programs to promote competent parenting should also focus on improving the social environment of parenting (Leyendecker et al. 2006). This should involve structural issues of poverty and inequality or the comprehensive meaning of family support that ensures that the family is supported in various aspects, not just when it is in crisis (Dolan et al. 2020). Overall, in adopting and implementing the comprehensive family support programs, the state itself, as Lakoff (2004, 2009) suggests, should embody the Nurturant Parent Model.

The so-called geopolitization of human rights in recent years makes it necessary to use non-political concepts, such as parenting styles supported by child psychology and humanistic education, to defend the universality of children's rights, which is essential for human flourishing. We also agree with Hanson and Peleg's (2020, p. 31) view that scholars of all types and backgrounds are welcome to reflect on how analytical and conceptual tools can be strengthened to further enhance the theoretical sophistication of children's rights.

It is important to show that children's rights-friendly parenting is critical to ensuring that children grow up happy, healthy, and mature regardless of their cultural and geographic context. In summary, children's rights provide appropriate direction to parental authority and responsibility to realize the child's well-being and contribute to overall human flourishing.

**Author Contributions:** Conceptualization, G.Z. and Z.N.; methodology, G.Z. and Z.N.; writing— original draft preparation, G.Z.; writing—review and editing, G.Z. and Z.N.; supervision, G.Z.; project administration, G.Z.; funding acquisition, G.Z. All authors have read and agreed to the published version of the manuscript.

**Funding:** This research was funded by the research grant provided by the Committee of Science, Ministry of Higher Education and Science, Republic of Kazakhstan, grant number is AP08856467. The APC was funded by the same grant.

**Institutional Review Board Statement:** Not applicable.

**Informed Consent Statement:** Not applicable.

**Data Availability Statement:** Not applicable.

**Conflicts of Interest:** The authors declare no conflict of interest.

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
