# Peer review of "The Need to Bridge the Gap between Research on Children’s Rights and Parenting Styles: Authoritative/Democratic Style as an Acultural Model for the Child’s Well-Being"

_socsci, doi:10.3390/socsci12010022_

Round 1

Reviewer 1 Report

I enjoyed reading this and many interesting points are outlined. Whilst the style of written English is generally good there are many points where revision is required. I would argue that there are two articles here - the single article is over-ambitious.  One article is an excellent discussion / think piece about children's rights and parenting. You write well on this topic and you are very well-informed. (In addition you should see Dolan et al 'Family support as a right of the child' in Social Work and Social Review 21(2) and the related work of the eurofamnet.eu). This would be a article worthy of publication in its own right. The second article could focus on the student work. We need to know much more about this. We need to know about the characteristics of the students; age, gender, how many are parents and so on. We also need to know more about methodology: how were the papers analysed and more about the major emerging themes. Also what were the assessment questions and the learning outcomes? Addressing these issues would make an interesting research paper. Good luck in developing your work. 

Author Response

My co-author and I have analyzed the comments of our reviewers with great interest.

We would like to thank our reviewers very much, whose valuable and collegial comments and suggestions helped us to make our arguments clearer and more accurate. We were also able to improve the style and enrich our references.

 In response to the reviewers' feedback and suggestions, we accept all suggestions.

Regarding the reviewer's comments about the potential to develop new research, we decided to write another article based on our empirical data in the future. We were unable to obtain written consent from the students and parents of young children with whom we interacted 

At this stage, we designed our research as a conceptual-normative article because [we hope] it will strengthen the position of human rights activists and progressive groups in Kazakhstan and neighboring countries who are working to get parliament to draft a new law prohibiting all types of violence against children and women. This draft law was put on hold due to fierce opposition from some parents' groups in 2019-2020 who were misled by false information (we mentioned this in the revised version) 

 We also added the concepts of the child's well-being, and human flourishing.   

1.We carefully rewrote and extensively edited the manuscript.

2.Our colleague from Australia (a professional user of academic English) thoroughly read the final revised version of the article 

3.We clarified our research paradigm (critical realism), the type of our article/research, and the research questions. We tried to build our arguments on the findings of critical realism's insights.

We have made it clear in the revised version that our article is conceptual-normative and incorporates the ideas of critical realism.

4.We have rewritten the introduction and conclusion and added new subsections, 

5. We have expanded the references and used the recommended sources. We have benefited much from Dolan et al. (2020). Family Support as a right of the child.

Thank you very much for reviewing our draft!  

Reviewer 2 Report

The topic and approach of this article are innovative. But the scientific approach is not very strong with regard to referencing to experience and students' attitudes without a thorough empirical study and empirical data. The main aim and conclusion of this publication are mainly based on the author's personal experience and recommendation to approach Eurasian countries with a strong paternalistic approach with an integration of parenting styles and relevant children's rights. Chapter 3 needs revision; there is repetition (see par. 2 and 3 of chapter 3), chapter 3 is repetitive also with other chapters of the article. In chapter 5 (analysis and integration of children's rights and parenting styles) any elaboration on ' the evolving capacities of the child' is missing; this should be added. In chapter 6 (par. 3) information is given about students' perceptions, but a clear and thorough approach is lacking; I would suggest to nuance this par. because the wording is now too strong for the data/sources used. Lastly, a thorough language check is needed, some sentences are not clear or incorrectly formulated.

some minor details:

- p. 12 par. 4, last sentence: not correct. clear. par. 4 sentence 1: authoritarian is not correct, should be authoritative

Author Response

My co-author and I have analyzed the comments of our reviewers with great interest.

We would like to thank our reviewers very much, whose valuable and collegial comments and suggestions helped us to make our arguments clearer and more accurate. We were also able to improve the style and enrich our references.

 In response to the reviewer's feedback and suggestions, we accept all suggestions.

1.Regarding the reviewer's comments about:

-“scientific approach is not very strong with regard to referencing to experience and students' attitudes without a thorough empirical study and empirical data”.

-“Concerning information is given about students' perceptions, but a clear and thorough approach is lacking” —  at this stage, we have decided to write another article in the future based on our empirical data. We were not able to obtain written consent from the students and parents of young children with whom we interacted. 

Therefore, we use the data from our experience in this article only as "anecdotal" at this state in this article. And we have omitted most of the information and comments about our empirical research (previous chapter 6, in the revised version chapter 7) 

We designed this draft (ongoing research) as a conceptual-normative article because [we hope] it will strengthen the position of human rights activists and progressive groups in Kazakhstan and neighboring countries who are working to get parliament to draft a new law prohibiting all types of violence against children and women. This draft law was put on hold due to fierce opposition from some parents' groups in 2019-2020 who were misled by false information (we mentioned this in the revised version) 

2.We have carefully rewritten and extensively edited the manuscript;

3.Our colleague from Australia (a professional user of academic English) thoroughly read the final revised version of the article; 

4.We clarified our research paradigm (critical realism), the type of our article/research, and the research questions. We tried to build our arguments on the findings of critical realism's insights.

We have made it clear in the revised version that our article is conceptual-normative and incorporates the ideas of critical realism.

5.As recommended by our reviewer, we rewrote Chapter 3, including the conclusion, and added new subsections,

6. In Chapter 6 (previously 5), we addressed the evolving capacities of the child.

7.We deleted the repetitive paragraphs and sections with typos.

8. Finally, we have expanded the references.

Thank you very much for reviewing our draft!